# Research on the Adhesive Performance of a Biomimetic Goat Hoof Track Shoe Pattern

**DOI:** 10.3390/biomimetics7020080

**Published:** 2022-06-14

**Authors:** Fu Zhang, Chaochen Zhang, Shuai Teng, Xiahua Cui, Shaukat Ali, Xinyue Wang

**Affiliations:** 1College of Agricultural Equipment Engineering, Henan University of Science and Technology, Luoyang 471003, China; 190319041067@stu.haust.edu.cn (C.Z.); 180318040209@stu.haust.edu.cn (S.T.); 200320041498@stu.haust.edu.cn (X.C.); 210321041649@stu.haust.edu.cn (X.W.); 2Henan International Joint Laboratory of Intelligent Agricultural Equipment Technology, Luoyang 471003, China; 3Collaborative Innovation Center of Machinery Equipment Advanced Manufacturing of Henan Province, Luoyang 471003, China; 4Wah Engineering College, University of Wah, Wah Cantt 47040, Pakistan; shaukat.ali@wecuw.edu.pk

**Keywords:** bionic, goat hoof, macrostructure, micro-morphology, track shoe pattern, adhesive performance

## Abstract

In this paper, reverse engineering technique was employed to extract the ridges of the hoof ball contour, and hoof ball tissue structure was analyzed based on the bionic prototype of goat hooves. The quantified geometric features were used to design the bionic track shoe pattern, which can enhance its adhesive performance and solve the problem that agricultural tracked vehicles in hilly and mountainous areas are prone to slip due to poor adhesive performance. The monolithic structure of the biomimetic goat hoof track shoe pattern and the ordinary one-line track pattern were arranged and combined; they included six kinds of track shoe models and the adhesive performance was compared. A discrete element system was established based on soil parameter determination to compare the maximum adhesion of different track shoe models. The bionic track shoe samples were prepared for soil bin tests to verify the reliability of the discrete element analysis results. Compared with the ordinary track shoe, the adhesion of the optimal bionic track shoe was improved by 9.1%.

## 1. Introduction

Tracked vehicles have better traffic capacity under soft ground conditions in hilly areas. The engine power of agricultural tracked vehicles commonly used in this area is high, and its power performance meets the traction requirements, while the adhesive performance of tracked vehicles to soft ground is weak, and it is easy to slip in the process of driving, climbing, obstacle avoidance and ditch crossing [1,2], which seriously restricts the passability of tracked vehicles. Therefore, it is important content for the research and development of agricultural tracked vehicles in hilly and mountainous areas to improve the adhesion of tracked vehicles to the ground.

As the only part of tracked vehicle in direct contact with the ground, the structure and pattern of track plate are closely related to its strong adhesion and high traction efficiency [3,4]. Orthogonal experiments have been used to study the influence of parameters on the adhesion, such as the width and height of the track shoe pattern [5]. The mathematical model of adhesion was used by Wu et al. [6] to study the relationship between the track shoe and the adhesion of the collector, and found that the adhesion was the strongest when the tooth height was 15 cm. Another study analyzed the theory of the discrete element model of the track shoe adhesive performance and found that the track shoe adhesion force, which decreased with the rearward arrangement of the track shoe patterns, was related to the adhesion perpendicular to the driving direction [7]. The Baker method was used to analyze the influence of track shoe parameters on the traction performance of tracked vehicles in another study and found that the height and thickness of the track shoe pattern were closely related to the traction performance of the vehicle [8]. To solve the problem of the track shoe of the deep-sea mining concentrator walking and slipping on the seabed, the relationships between the maximum traction force and the structural parameters of the track shoe based on bionic theory were analyzed by [9], and the optimal running speed of 629.72 mm/s was obtained. The influence of soil moisture content and pattern height on the traction force for a single pattern was studied to obtain the optimal combination parameters. This study found that Single Grouser Shoe (SGS) with a 1 cm grouser height was better when the moisture content (MC) was lower than 15%. The optimum grouser height increased with the increase in MC until the peak value of 26 cm at 36% MC [10]. In addition, the scholar further studied the material properties and compared the traction performance of steel track shoe and rubber track shoe under different soil moisture contents. The results showed that when other conditions remain the same and the soil moisture content was greater than 15%, the traction performance of the steel track was better than that of the rubber track [11]. Focusing on the adhesion problem between black soil and tracked vehicles in northeast China, the figure-eight track shoe was obtained as the research object in another study. The height, thickness and opening angle of the track teeth were obtained as the test factors, and the adhesion force was selected as the test index. Then, the optimal combination was obtained after analysis [12]. In order to evaluate the tractive performance of the underwater unmanned crawler bulldozer, the mechanical properties of π-type, T-type and V-type track shoe were analyzed. In another study, the coupling mechanism between the structural parameters and the traction performance was obtained by changing the length, width and height of the pattern, which provided a dynamic basis for the anti-skid and stability of the unmanned control crawler bulldozer under the water [13]. The rheological calculation formula of the traction force for the actual pattern structure has been deduced on the basis of the pressure-shear rheological model of the deep-sea sediment simulator, and a multi-objective structural optimization model was constructed to determine the optimal track shoe pattern structure parameters. It could provide theoretical basis for the safe operation and optimal design of crawler mining vehicles [14].

Some researchers have found that the hoof ends of goats’ limbs have evolved into irregular shapes that can adapt to different ground conditions, and the hoof end structure changes dynamically when interacting with the ground conditions, and they can move on complex ground conditions such as uneven or soft ground conditions [15,16], showing strong adhesion and passing ability, which can provide bionic inspiration for the design of bionic track shoe patterns.

With the deepening of research, some researchers have found that the rigid–flexible coupling structure of goat hooves plays a key role in the movement. Abad et al. [17,18] recorded the force of the hoof, the position of the hoof and the effect of the three joints within the hoof on the requirement for motor slip through experiments on the fabricated bionic cloven hoof. The surface morphology, structure and material composition of goats have been studied, and it was verified that the surface morphology and microstructure had different effects on slip resistance and shock buffering through the finite element analysis [19]. Inclined holes with a 55° inclination angle inside the goat hoof box have been observed, ands used as a bionic prototype to design and optimize a bionic unit. The performance test results showed that the damping effect of the bionic damper unit was better than that of the non-porous samples [20].

The existing research shows that the track shoe adhesive performance is mainly improved by changing the design of the track shoe pattern structure. However, few studies have considered the application of natural biological features in this context. Although many scholars have carried out research on goat hooves, there are few reports on the research that considers the design of the contour curve of the macroscopic characteristic part of the organism and its tissue structure characteristics.

Therefore, this study attempts to combine the hoof ball profile with strong adhesion characteristics with its tissue structure to design a track shoe pattern structure, and analyze it based on the discrete element method (DEM).

## 2. Materials and Methods

### 2.1. Bionic Structure Design

#### 2.1.1. Macro Contour Extraction

The previous study by the research group found that goats had better rear hoof adhesion under the condition of uphill movement [15,16]. Therefore, the left rear hoof of a 3-month-old male domestic Boer goat was selected as the research object in this paper.

The geometric model of the goat’s hoof (Figure 1a) was obtained by computed tomography (CT) scanning and reverse reconstruction technology, then the ridge line of the hoof ball was projected by CATIA. The coordinates for the ridge line of the goat hoof ball (Figure 1b) were extracted through the processes of grayscale, binarization and point drawing.

Finally, the curve fitting tool in the Matlab was used to extract the point coordinates and perform polynomial fitting to obtain a curve equation such as the formula:(1)$$f(x)=0{.00007298x}^{3}-0{.01195x}^{2}+0.1134x+78.29,\;10\;\leq \;x\;\leq \;158,\;\mathrm{mm}$$

The root mean square error of the equation is RMSE = 1.504, and the fitting coefficient *R*^2^ = 0.9851.

#### 2.1.2. Observation on Tissue Structure of the Goat Hoof Ball

The histological observation of the claw ball specimens was performed with a stereo microscope (Phenix, XTL-165-LT) after the HE staining technique (as shown in the Figure 2). The goat hoof ball presents a multi-layered tissue structure, which is composed of three layers of stratified epithelium, dermis and subcutaneous tissue from outside to inside.

As shown in Figure 2, in order to describe the spatial hierarchy for the parts of the goat hoof ball, the bottom of the goat hoof was regarded as the xoy plane, and the direction from the plane to the bottom of the goat hoof was the positive direction of the z-axis.

It was found that the hoof ball has an elliptical hole-like structure near the meat hoof and was arranged linearly, its spacing was between 0.068~0.089 mm, and the ratio of transverse to longitudinal axis was about 2:1. When the structure was subjected to impact (when it was subjected to force along the z-axis), it deformed due to the contact area with the ground being increased and hence enhanced its adhesion with the ground. This can provide the basis for the three-dimensional design of the bionic track shoe pattern.

#### 2.1.3. Structure Design of Bionic Track Shoe Pattern

The goat hoof ball outline has good adhesive performance due to the soil-cutting and soil-fixing effect, which was used as the bionic inspiration to design the bionic track shoe pattern. Based on the structure of the goat hoof ball, bionic grooves were arranged on the bionic curved surface to realize the design of the goat hoof track shoe pattern monomer with the combination of macro profile and micro morphology.

The specific design process for the bionic pattern surface was as follows: the bionic curve fitting equation and its value range obtained in Section 2.1.1 were entered into the Solidworks software, and the corresponding contour curve was obtained. Considering the integrity of pattern parameters, the interval of *x* was rounded, and the bionic curve meets the following equation:(2)$$f(x)=0{.00007298x}^{3}-0{.01195x}^{2}+0.1134x+78.29,\;10\;\leq \;x\;\leq \;160,\;\mathrm{mm}$$

In addition, considering the dynamic ground contact process for the track pattern to the vehicle in the actual driving process, the bionic curve was functionally divided into three parts: soil-cutting section, support section and soil fixation section, so as to enhance its dynamic adhesive performance. Among them, the range of soil consolidation section was 0 ≤ *x* ≤ 70, the range of support section was 70 < *x* ≤ 130 and the range of the cutting section was 130 < *x* ≤ 160. After it was closed, the bionic pattern section was obtained. Combining with the actual track pattern size, without affecting its functional effect, the size of the pattern cross-section sketch was reduced to 1/2 of the original size; the bionic track shoe pattern cross-section sketch is shown in Figure 3. Then, it was stretched along the length direction of the pattern body with a stretching width of 300 mm.

On the basis of the structure of the biomimetic goat hoof track shoe pattern shown in Figure 3, the bionic groove was designed based on the microscopic elliptical structure of the goat hoof ball. For the pattern width, the short axis of the ellipse is b = 2 mm, then the long axis is a = 4 mm. The center of the ellipse is located on the bionic curve, which divides the above-mentioned ellipse structure into two parts, and retains part of the curve close to the pattern body.

The single structure of the biomimetic goat hoof track shoe pattern with grooves is shown in Figure 4.

Nine bionic grooves were arranged on the body of the single pattern, and the center of the first bionic groove was arranged at the vertex of the curve interval of the biomimetic goat hoof track shoe pattern. The rest of the bionic grooves were arranged in sequence according to the spacing. The horizontal spacing of the bionic groove structure in the solid soil section curve, the support section curve and the soil-cutting section curve were L1, L2 and L3, respectively, with spacing ratio of L1:L2:L3 = 5:3:2. According to the pattern size designed in this paper, L1 = 12.5 mm, L2 = 7.5 mm and L3 = 5 mm.

#### 2.1.4. Overall Structure Design of Bionic Track Shoe

In Section 2.1.3, two kinds of bionic track shoe pattern structures were designed, and the ordinary I-shaped track shoe pattern was selected as a comparison. On the premise of ensuring the same cross-sectional area and height of the pattern, the width of the conventional pattern for comparison was calculated as 51 mm. The patterns were categorized and represented by alphabetic codes for testing as shown in Table 1.

However, in the actual application scenario, the track shoe was composed of a plurality of track patterns. In order to compare the differences in the adhesion performance of the three patterns, it was necessary to connect and combine the different patterns through the track shoe matrix. The designed track shoe matrix size is Length × Width × Height = 300 mm × 300 mm × 30 mm, and two track pattern monomers are arranged on each track plate, and the spacing is set to 100 mm. Considering only the one-way movement of track shoes, the test scheme of pairwise combination of different types of patterns is shown in Table 2.

### 2.2. Soil Parameters Measurement

#### 2.2.1. Moisture Content and Density

The volume of soil was obtained by collecting soil samples with a ring knife, and the weight of soil was measured by Leqi LQ-C3003 electronic balance (accuracy 0.001 g). After repeating 5 times, the density of soil was calculated to be 1330 Kg/m^3^.

In this paper, the drying weight measurement method was used to measure the soil moisture content, and Equation (3) was used to calculate the soil moisture content. The average soil moisture content was calculated to be 10%.
(3)$$\omega =\frac{m_{1}-m_{2}}{m_{2}}\times 100\%$$ where $m_{1}$ and $m_{2}$ are the weights of wet soil and dry soil (g), respectively.

#### 2.2.2. Poisson’s Ratio

In this experiment, the soil with 10% moisture content was used as the raw material, and the sample was prepared by a ring knife. The Poisson’s ratio was measured by a texture analyzer (model: TA. XTC-16; accuracy: 0.01 g; displacement accuracy: 0.001 mm; data acquisition frequency: 500 groups/s), which was used as a loading device to conduct a compression deformation test on soil samples, and the height and diameter of soil samples should be measured before the test. Poisson’s ratio of the sample was calculated according to Equation (4). The average result of five experiments in this study was 0.33.
(4)$$\mu =\left|\frac{\varepsilon_{x}}{\varepsilon_{y}}\right|=\frac{\Delta D/D}{\Delta H/H}$$
where *μ* is the Poisson’s ratio, $\varepsilon_{x}$ is the radial strain of the sample; $\varepsilon_{y}$ is the longitudinal strain of the sample, ΔD is the absolute radial deformation of the sample (mm), D is the original diameter of the sample (mm), ΔH is the absolute longitudinal deformation of the specimen(mm) and *H* is the original longitudinal height of the sample (mm).

#### 2.2.3. Elastic Modulus

In the test, the height *H* of the sample before compression was measured by digital vernier caliper, and then it was naturally placed on the platform of universal tensile machine (Changchun Institute of Mechanical Science Co., Ltd., Changchun, China, DNS02, accuracy grade 0.5). The load was applied to the sample at a loading speed of 5 mm/min. The soil was deformed after being pressed, and the data of force (F)-deformation (ΔH) were recorded. The above process was repeated, and the elastic modulus of the sample is calculated as 1 × 10^7^ Pa according to Equations (5) and (6).
(5)$$E=\left(\frac{F}{A}\right)/\varepsilon$$
(6)$$\varepsilon =\frac{\Delta H}{H}$$
where *E* is the Elastic modulus(Pa), *F* is the axial load on specimen (N), *A* is the contact area (mm^2^), *ε* is the strain, ΔH is the absolute longitudinal deformation of the specimen (mm) and *H* is the original longitudinal height of the sample (mm).

### 2.3. Soil Repose Angle Test

#### 2.3.1. Measurement Results of Soil Repose Angle

Repose angle is an important basic data of soil, and there are many complex movements of soil particles in the process of forming the repose angle, which can intuitively reflect the characteristics of soil flow, scattering, friction and bonding. In this paper, the bottomless cylinder-lifting method was used to verify the stacking angle test [21,22], the test device is shown in Figure 5.

Before the test, the soil with a moisture content of 10% was prepared, and the sundries and large blocks in the soil were removed. Then, the lower end of the stainless-steel cylinder was pressed against the loading platform by a universal tensile tester, and the stainless-steel cylinder was filled with soil from the upper end. Through the preliminary test, it was found that it was easier for the soil to form a stable slope when the cylinder was lifted at a speed of 50 mm/min. When all the soil was still, the right side of the mound was photographed vertically with a camera.

In order to reduce the error of manual measurement, the image processing method was adopted to process the soil repose angle image obtained from the experiment. The specific extraction process for the soil repose angle profile is shown in Figure 6.

The specific processing steps are as follows: firstly, the original soil repose image was grayed by Matlab. Secondly, an appropriate threshold was selected to binarize the grayed image. Image morphology was used to etch and expand to fill holes in binary images. Then, a complete edge contour curve was obtained by using the “bwperim” function in Matlab. Then, the image digitization tool in Origin software was used to convert the obtained edge contour curve into coordinate data and fit it linearly. Finally, the slope obtained by linear fitting was converted into the angle [23], which was the soil repose angle.

The above process was repeated five times, and the repose angle of the test was 27.37°.

#### 2.3.2. Simulation and Calibration of Soil Parameters

EDEM is widely used in soil particle research because of its rich functions [24]. There are many kinds of material models in the particle material database (Generic EDEM material model database, GEMM) built in EDEM. According to the measured parameters such as soil particle size, density and repose angle, the range of contact parameters between soil particles was obtained. The repose angle was used as the test evaluation index, and the Box–Behnken optimization method in Design-Expert11 was used for the calibration test. The test factor codes are shown in Table 3.

The simulation test for the soil repose under the same contact parameter (Figure 7) was repeated twice. After the particles were still, images were captured from two different directions of the generated mound, and the soil repose angle was measured by protractor function in the post-processing module of EDEM, and the average value was obtained. The simulation test results are shown in Table 4.

According to the test results in Table 4, the second-order regression model of the soil particle repose angle θ and three independent variables (coded values) was established by using Design-Expert, and the quadratic polynomial equation was obtained as follows:(7)$$\begin{array}{l} \theta =26.26+9.02A+0.65B-0.525C+0.05AB\\ -0.05AC-1.45BC-9.91A^{2}-0.555B^{2}-0.255C^{2} \end{array}$$

The determination coefficient *R*^2^ of the regression equation was 0.9711, and the correction determination coefficient *R*^2^_adj_ = 0.9339. Both values were close to 1, which indicates that the fitting equation has high reliability and practical significance. The coefficient of variation, CV = 10.13%, and accuracy = 13.2842, indicate that the model has good accuracy and reliability.

The experimental regression model in Table 4 was analyzed by variance analysis, and the results are shown in Table 5. 

The results show that *p* = 0.0001 < 0.01, indicating that the regression model is very significant and can be used to predict the soil repose angle. The simulated primary term A has a very significant effect on the accumulation angle, and the quadratic term *A^2^* has a very significant effect on the repose angle.

By using the optimization function of Design-Expert, several groups of solutions were obtained, and the optimal solutions closest to the measured data were selected: the soil–soil restoration coefficient was 0.63, the soil–soil dynamic friction coefficient was 0.2 and the soil–soil static friction coefficient was 0.3. Under the reorganization optimal solution, the simulation value of the soil repose angle is 27.70. The relative error between the soil repose angle and measured value was 1.2%. The comparison between the measured test and the simulation test is shown in Figure 7.

The track shoe material in the simulation model was selected as rubber. The simulation parameters for the soil particles and geometry in the discrete element method are shown in Table 6.

### 2.4. Contact Model

The contact model was used to describe the contact behavior of particles in contact with each other. Whether the simulation results are of reference significance and can reflect the actual working conditions have a lot to do with whether the particle contact model selected in the simulation process is consistent with the actual soil. There are many contact models built into EDEM2020, for example, the default Hertz–Mindlin model, but this model only considers the elastic deformation of particles, but ignores the bonding force between particles [25]. The Hertz–Mindlin model combined with the JKR (Johnson–Kendall–Roberts) model, referred to as the JKR model and adds the JKR surface energy theory to the Hertz–Mindlin model to describe the viscosity between particles, is a cohesive contact model [26]. The Hertz–Mindlin model combined with the bonding (HMB) model adsorbs particles together to form large pieces of materials through bond bonds. After bond bonds are produced, the normal force $F_{n}$, tangential force $F_{t}$ and torque $M_{n}$ and $M_{t}$ given to soil particles by the outside are calculated according to Equations (8)–(13). When the bond between soil particles breaks under the action of external force, these bulk materials will be reduced to particles and exist alone [27,28]. According to the type of soil used in the experiment, the Hertz–Mindlin model with the bonding model in EDEM2020 is selected as the contact model among soil particles [29].
(8)$${\delta F}_{n}=-v_{n}S_{n}{A\delta }_{t}$$
(9)$${\delta F}_{t}=-v_{n}S_{t}{A\delta }_{t}$$
(10)$${\delta M}_{n}=-\omega_{n}S_{t}{J\delta }_{t}$$
(11)$${\delta M}_{t}=-\omega_{t}S_{n}\frac{J}{2}\delta_{t}$$
(12)$${A=\pi R}_{B}{}^{2}$$
(13)$$J=\frac{1}{\;2}{\pi R}_{B}{}^{4}$$
where *A* is the contact area between soil particles (m^2^), *R_B_* is the bond radius between soil particles, *J* is the polar moment of inertia of section (m^4^), $S_{n}$ is the normal stiffness of bonded particles (N/m), $S_{t}$ is the tangential stiffness of bonded particles (N/m), $v_{n}$ is the normal component of particle velocity (m/s), $v_{t}$ is the tangential component of particle velocity (m/s), $\omega_{n}$ is the normal component of particle angular velocity (rad/s), $\omega_{t}$ is the tangential component of particle angular velocity (rad/s) and $\delta_{t}$ is the time step (s).

The related parameters between soil bond–bond refer to the reference [30,31]: normal contact point stiffness, 1 × 10^6^ Pa/m^3^, tangential contact point stiffness, 1 × 10^6^ Pa/m^3^, critical normal stress, 10,000 Pa, critical tangential normal stress 10,000 Pa and bond radius 1.3 mm.

### 2.5. Establishment of Simulation System

The numerical simulation analysis of the designed bionic track shoe was carried out by using EDEM2020 to study the influence of different pattern combination schemes on the adhesive performance of the track shoe. The rationality and superiority of bionic track shoe pattern design were verified. Soil properties, material properties and other parameters in the simulation system are shown in the Table 6. When the diameter of soil particles was set to 2 mm, a total of 320,000 particles were generated, and the size of the particle bed was: Length × Width × Height = 750 mm × 600 mm × 200 mm.

The track plate coordinate system was adjusted to be consistent with the coordinate system in EDEM and imported into the EDEM2020, as shown in Figure 8.

The specific simulation steps were as follows:(1)The track plate model was set to fall into the soil particle bed at the speed of 8 mm/s until the track pattern was completely submerged into the particle bed.(2)After resting for a period of time, the track plate was set to move horizontally at the speed of 8 mm/s.

### 2.6. Preparation of Track Shoe

In this study, natural rubber plate (Length × Width × Height: 300 mm × 300 mm × 30 mm) was selected as the processing raw material of track plate matrix. The imitating goat hoof track shoe pattern contains bionic curves, bionic grooves and other complex structures, which is not easy to be processed by conventional methods, so it was processed by 3D printing (J G Maker, A6). Considering the problem of material properties, thermoplastic polyurethanes (TPU) soft materials similar to rubber materials were selected to print. The six test track shoes after processing are shown in Figure 9.

### 2.7. Soil Bin Test

The test was carried out in a self-designed soil bin, and the electronic universal testing machine of Section 2.2.3 was used as the power source and data acquisition device to provide the power of the test track shoe and the test data acquisition work. When measuring the traction force, the soil bin was placed in front of the testing machine, the vertical tensile force was converted into the horizontal tractive force by the fixed pulley and the track shoes in the soil bin were pulled to move in the horizontal direction. The track shoe test system is shown in Figure 10.

## 3. Results and Discussion

### 3.1. Numerical Analysis of Simulation Results

The six combination schemes shown in Table 2 is simulated according to the settings described in Section 2.5. After the completion of simulation, the force on the *x*-axis direction of each model was derived through the post-processing function of the software and drawn onto the same figure, as shown in Figure 11. As can be seen from the figure, the variation trend of the adhesive force of six kinds of track shoe models is basically the same. The force of the six track shoe models increased sharply to the maximum value in a short time, which is, the maximum adhesion under the condition of non-slipping. Then, the track shoe continued to shear the soil and the adhesion began to decrease. With the increase in traction displacement, the soil accumulated in front of the track shoe was increased, and the curve showed an upward trend. To ensure the accuracy of the test results, the data with large fluctuations at the beginning of the test and the data after the soil accumulation at the back were eliminated, and the force data within 8 ~ 48 mm were selected for analysis.

The maximum adhesive force of each track shoe is shown in Table 7. Compared with the ordinary track shoe, the adhesive performance of the track shoe with bionic pattern was improved, and the combination scheme CC had the best adhesive performance, which was 9.1% higher than AA. This proves the rationality for the design of the bionic track shoe pattern.

### 3.2. Microanalysis of Simulation System Based on EDEM

Through numerical analysis, it was found that in the simulation system, the adhesive performance of CC with bionic pattern was the best, which was better than that of ordinary track shoe. In order to further explore the internal adhesion mechanism of the imitating goat hoof track shoe pattern. Scheme CC and AA were selected as the research objects, the soil particle flow field and contact force field were analyzed from the microscopic changes through the intuitive post-processing function of EDEM. The comparison of the particle flow field formed by the two kinds of track plates is shown in Figure 12 and Figure 13.

In the figure the size of the arrow represents the velocity and the arrow direction represents the velocity direction. It can be seen from the figure that the soil particle flow field changes with the movement of the track. The particle flow field formed by the conventional track plate has a large range, but it was not dispersed and could not form a firm reverse force, but the bionic pattern shape of the bionic track shoe makes the particle flow field form a fixed direction. The “self-locking” phenomenon is formed to provide greater reverse force, which is helpful to enhance the adhesion between the bionic track shoe and the ground.

In this paper, a 500 mm × 500 mm area was selected as the geometric center of the track shoe. Comparing the proportion distribution of different force particles between bionic track shoe and conventional track shoe in this area is shown in the Figure 14 and Figure 15. In the stress range of 0.0016–0.0032 N, the number of particles under the conventional track shoe accounts for 42.02% of its total particles, while the number under the bionic track shoe accounts for 45.85% of its total particles, which indicates that under the same conditions, the number of particles with strong stress around the bionic track shoe accounts for a higher proportion.

### 3.3. Comparative Analysis of Simulation and Test Results

In order to compare the error between the maximum adhesion force obtained by DEM and soil bin test, the numerical simulation results for each track shoe and the verification results for the soil bin test were drawn in the same diagram. The comparison of the tractive force between the simulation and the test is shown in Figure 16.

It can be seen from the figure that the fluctuation of the simulation result is smaller than that of the soil bin test. The reason is that the soil environment in the experiment was more complex and there were artificial errors, but the simulation results can be controlled in a certain range, which is not much different from the test results. The traction error between the simulation and the soil bin test is shown in Table 8.

## 4. Discussion

Our team found that the ridge line of the goat hoof ball plays an important role in its movement, combined with the geometric ellipse structure in the tissue structure of this part, which provides a new idea for the design of bionic track shoe pattern.

In the simulation environment and soil groove experiment constructed in this paper, the track shoe with bionic track shoe pattern has a good adhesive performance, and the track shoe pattern with bionic groove has the best adhesion performance to the ground, which is of great significance for studying the development of agricultural machinery and agricultural bionic parts in hilly and mountainous areas.

In future, a complex slope environment may be set to explore the adhesive performance of bionic track shoe patterns, and the influence of material properties on the wear rate should be further studied.

## 5. Conclusions

This study was aimed at addressing the problem of poor adhesion of tracked vehicles in hilly and mountainous areas; based on the macro profile curve and microstructure of the goat hoof ball, the imitating goat hoof track shoe pattern was designed. Numerical simulation analysis of the bionic track shoe was carried out, and the simulation results were verified by the self-built track shoe test system. Through the above research, the following conclusions are drawn:(1)The contour curve of the goat hoof ball ridge line was extracted, and its mathematical model was constructed. Combined with the structure of hoof ball, bionic groove was designed, and based on both, the single structure of the goat hoof track pattern was designed.(2)The soil moisture content, density, Poisson’s ratio, elastic modulus and other parameters were measured. Taking the measured results for the soil stacking angle as the optimization target value, the Box-Behnken optimization method was used to obtain the optimal combination results for the soil contact parameters: recovery coefficient (0.63), static friction coefficient (0.2) and rolling friction coefficient (0.3), and the track shoe–soil simulation system was built.(3)Compared with the ordinary straight track pattern, the designed mountain-like sheep hoof track pattern has strong adhesion. The EDEM verification experiment showed that the change in the simulation value of the established track shoe-soil model is consistent with the trend of the experimental value, with the maximum error of 10.3%, which proves the reliability of the simulation model, and the adhesion performance of bionic track shoe is improved by 9.1% compared with that of conventional track shoe.

## Figures and Tables

**Figure 1 biomimetics-07-00080-f001:**
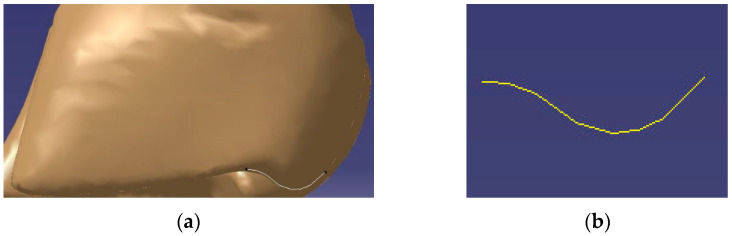
Biological prototype: (**a**) 3D model of a goat hoof and (**b**) bionic curve.

**Figure 2 biomimetics-07-00080-f002:**
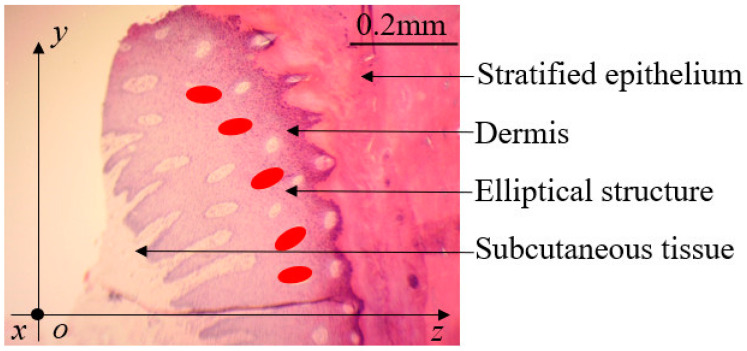
Observation of the tissue structure of the goat hoof ball.

**Figure 3 biomimetics-07-00080-f003:**
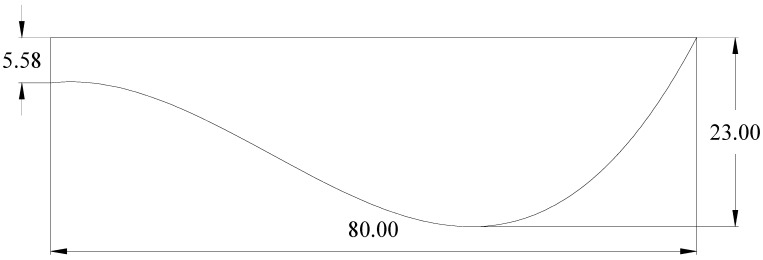
Biomimetic goat hoof track shoe section structure.

**Figure 4 biomimetics-07-00080-f004:**
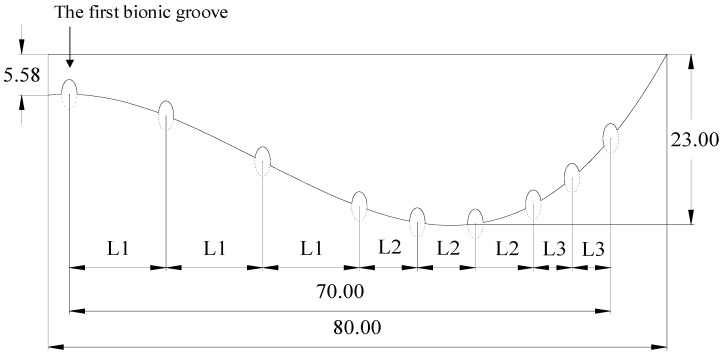
Structure schematic diagram of biomimetic goat hoof shoe with grooves.

**Figure 5 biomimetics-07-00080-f005:**
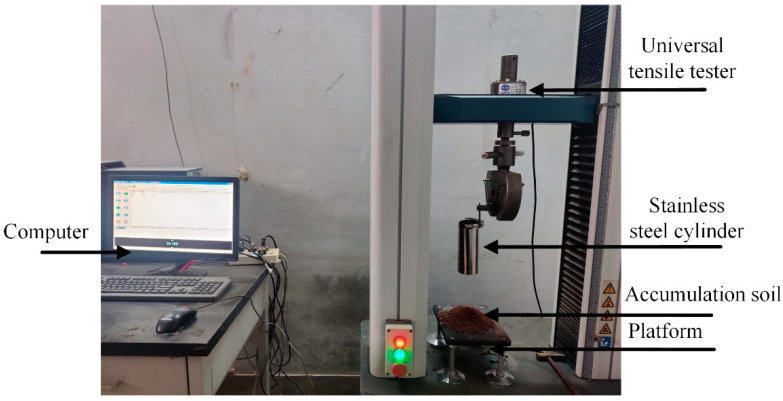
Soil repose angle test device.

**Figure 6 biomimetics-07-00080-f006:**
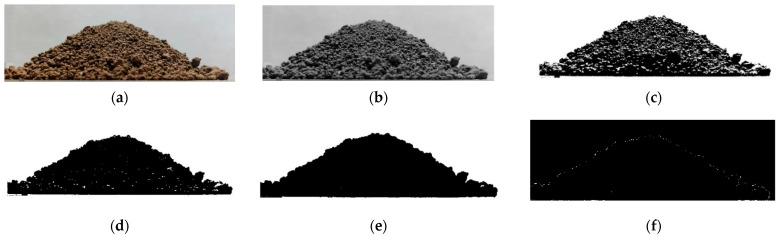
Edge contour extraction process for the soil repose angle: (**a**) original image, (**b**) grayscale image, (**c**) binary image, (**d**) before morphological treatment, (**e**) after morphological treatment and (**f**) contour image.

**Figure 7 biomimetics-07-00080-f007:**
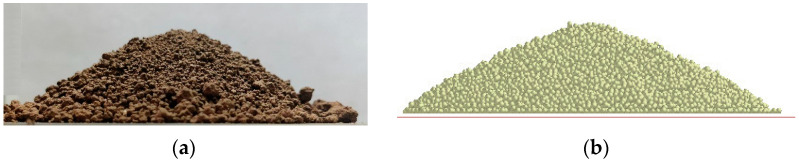
Comparison diagram of the soil repose angle between actual test and simulation test: (**a**) measured test results and (**b**) simulation test results.

**Figure 8 biomimetics-07-00080-f008:**
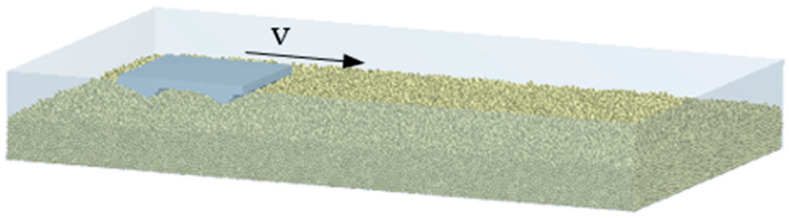
Track shoe–soil simulation system.

**Figure 9 biomimetics-07-00080-f009:**
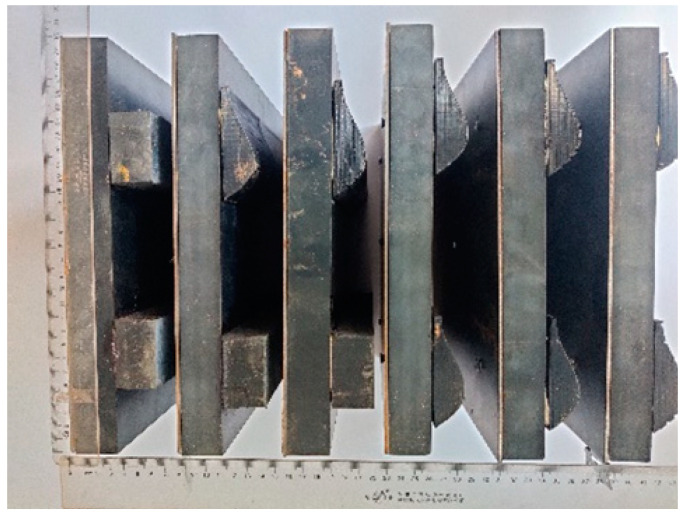
Specimens of track shoe.

**Figure 10 biomimetics-07-00080-f010:**
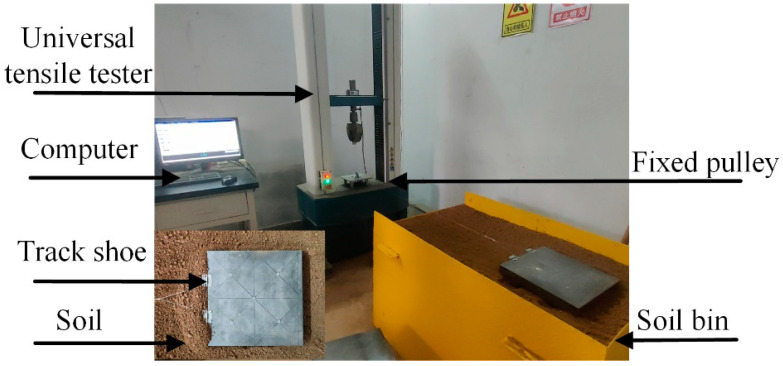
Track shoe test system.

**Figure 11 biomimetics-07-00080-f011:**
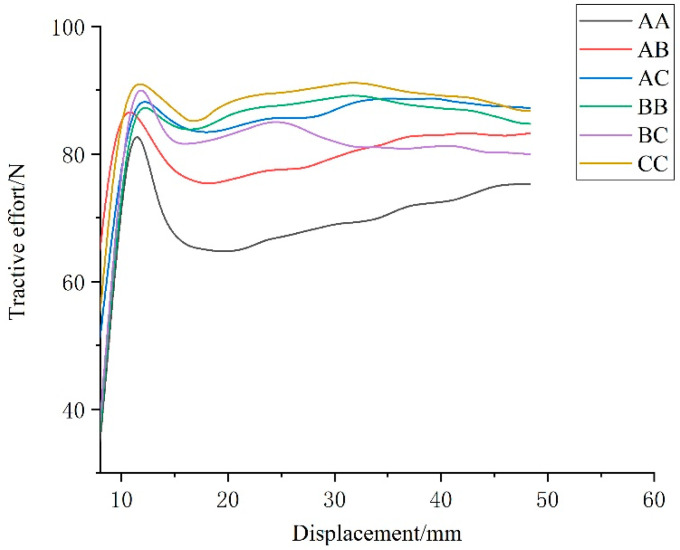
Tractive force during simulation of the six track shoes.

**Figure 12 biomimetics-07-00080-f012:**
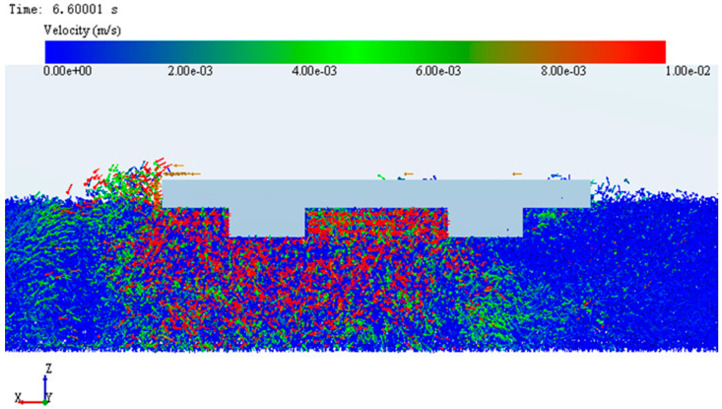
Particle flow field under ordinary track shoe.

**Figure 13 biomimetics-07-00080-f013:**
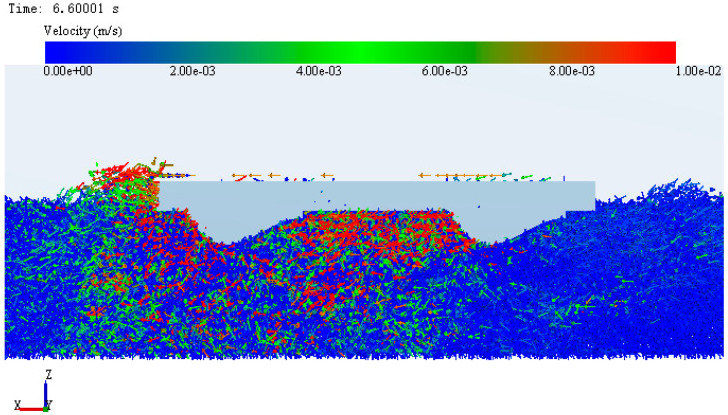
Particle flow field under bionic track shoe.

**Figure 14 biomimetics-07-00080-f014:**
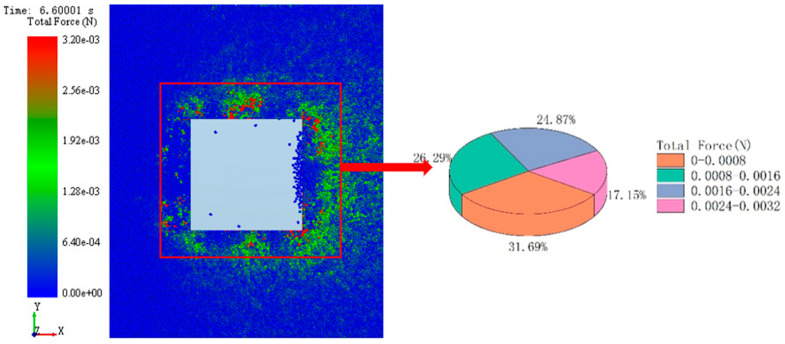
Stress distribution of particles under ordinary track shoe.

**Figure 15 biomimetics-07-00080-f015:**
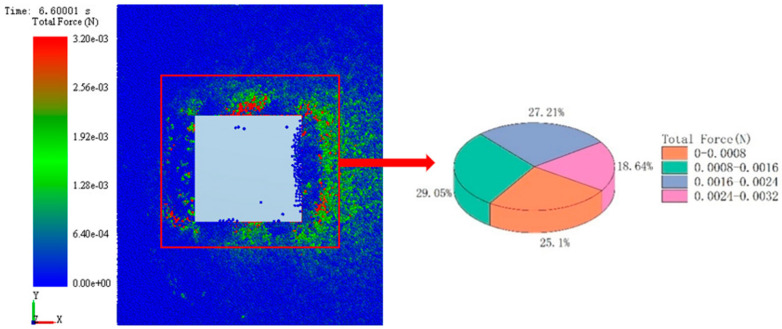
Stress distribution of particles under bionic track shoe.

**Figure 16 biomimetics-07-00080-f016:**
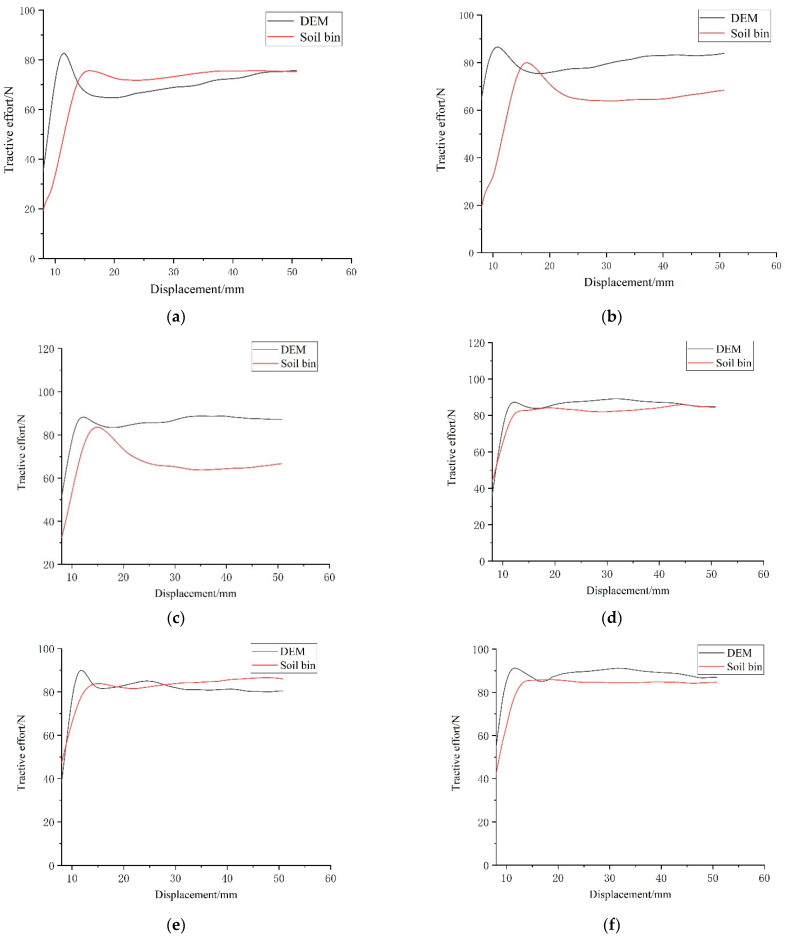
Comparison of the tractive effort-displacement between simulation and soil bin test: (**a**) AA scheme, (**b**) AB scheme, (**c**) AC scheme, (**d**) BB scheme, (**e**) BC scheme and (**f**) CC scheme.

**Table 1 biomimetics-07-00080-t001:** Category and size parameters for the code and test pattern.

Category	Code	Length × Width × Height (mm)
Ordinary I-shaped track shoe pattern	A	300 × 51 × 23
Biomimetic goat hoof track shoe pattern	B	300 × 80 × 23
Biomimetic goat hoof treads pattern with grooves	C	300 × 80 × 23

**Table 2 biomimetics-07-00080-t002:** Combination scheme of track shoe pattern.

Number	Combination Scheme	Schematic Diagram
1	AA	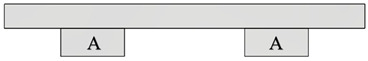
2	AB	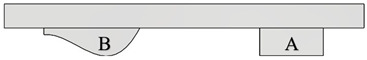
3	AC	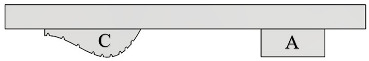
4	BB	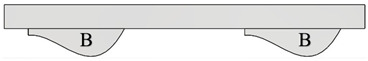
5	BC	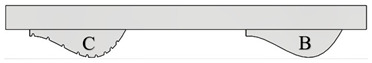
6	CC	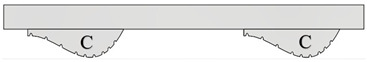

**Table 3 biomimetics-07-00080-t003:** Test factors and levels of soil repose angle calibration.

Code	Factor		
	Recovery Coefficient between Soil and Soil *A*	Dynamic Friction Coefficient between Soil and Soil *B*	Static Friction Coefficient between Soil and Soil *C*
−1	0.15	0	0.20
0	0.45	0.10	0.26
1	0.75	0.20	0.32

**Table 4 biomimetics-07-00080-t004:** Simulation test design of the soil repose angle.

Number	Factor			Repose Angle/(°)
	*A*	*B*	*C*	
1	0	−1	−1	23.8
2	1	1	0	24.6
3	0	0	1	24.1
4	0	0	0	29.1
5	0	0	0	28.9
6	−1	0	1	6.9
7	0	−1	1	24.5
8	0	1	−1	29.3
9	−1	−1	0	7.1
10	0	0	0	24.1
11	0	1	1	24.2
12	1	0	−1	25.4
13	0	0	0	25.1
14	1	−1	0	24.5
15	−1	1	0	7
16	−1	0	−1	6.7
17	1	0	1	25.4

**Table 5 biomimetics-07-00080-t005:** Analysis of variance of polynomial model of stacking angle Box–Behnken Test.

Variance Source	Sum of Squares	Degree of Freedom	Mean Square	F-Value	*p*-Value
model	1086.56	9	120.73	26.13	0.0001 *
*A*	651.61	1	651.61	141.03	<0.0001 **
*B*	3.38	1	3.38	0.7316	0.4207
*C*	2.21	1	2.21	0.4772	0.5119
*AB*	0.0100	1	0.0100	0.0022	0.9642
*AC*	0.0100	1	0.0100	0.0022	0.9642
*BC*	8.41	1	8.41	1.82	0.2193
*A* ^2^	413.09	1	413.09	89.41	<0.0001 **
*B* ^2^	1.30	1	1.30	0.2807	0.6126
*C* ^2^	0.2738	1	0.2738	0.0593	0.8147
Residual	32.34	7	4.62		
Lack of Fit	6.63	3	2.21	0.3438	0.7965
Pure Error	25.71	4	6.43		
Cor Total	1118.90	16			

**Table 6 biomimetics-07-00080-t006:** Basic parameters for the soil particles and geometry in the discrete element method.

Material Parameters	Value	Contact Parameter	Value
Soil density (kg/m^3^)	1330	Recovery coefficient between soil and soil	0.63
Rubber density (kg/m^3^)	960	Dynamic friction coefficient between soil and soil	0.2
Soil Poisson’s ratio	0.33	Static friction coefficient between soil and soil	0.3
Rubber Poisson’s ratio	0.45	Recovery coefficient between soil and rubber	0.61
Soil elastic modulus (Pa)	1 × 10^7^	Dynamic friction coefficient between soil and rubber	0.23
Elastic modulus of rubber (Pa)	3448	Static friction coefficient between soil and rubber	0.48

**Table 7 biomimetics-07-00080-t007:** Adhesive force comparison of different track shoes.

Number	Scheme	Mass (Kg)	Adhesive Force (N)	Rate of Increase/%
1	AA	7.18	83.6	—
2	AB	7.15	86.9	3.9
3	AC	7.10	88.7	6.1
4	BB	7.21	89.2	6.7
5	BC	7.19	90.5	8.3
6	CC	7.13	91.2	9.1

**Table 8 biomimetics-07-00080-t008:** Traction error analysis of simulation and soil bin test.

Number	Scheme	Result of Simulation/N	Result of Soil Bin Test/N	Error/%
1	AA	83.6	75.8	10.3
2	AB	86.9	80.1	8.6
3	AC	88.7	83.7	6.0
4	BB	89.2	84.2	6.0
5	BC	90.5	83.9	7.9
6	CC	91.2	85.9	6.2

## Data Availability

Not applicable.

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
