# Peer review of "Research on the Adhesive Performance of a Biomimetic Goat Hoof Track Shoe Pattern"

_biomimetics, 2022, doi:10.3390/biomimetics7020080_

Round 1

Reviewer 1 Report

  1. In chapter 2.1.1. Macro contour extraction, specify exactly which goat was used in the study (type, gender, age, wild or domestic, which leg).
  2. In formula (1), give the units of measurement.
  3. In Figure 1, specify the location of layers of stratified epithelium, dermis and subcutaneous tissue from outside to inside and elliptical hole-like structure
  4. Specify the units of measurement in Figure 3.
  5. In chapter 2.2.1. Moisture content and density, specify which soil was used. Where did you get this soil from?
  6. Give a citation to the description of EDEM software.
  7. Decipher the abbreviation of TPU.
  8. In chapter 2.6, display the mass of each of 6 tracked shoe specimens.
  9. In chapter 3.2, it is recommended to describe in more detail what is the difference between the AA and CC schemes from a physical point of view.
  10. It is necessary to comment on whether the resulting difference between the AA and CC schemes will remain if the movement is carried out in the opposite direction.
  11. It is necessary to discuss the practical significance of the obtained results. How will the resulting benefit compensate the complexity of manufacturing the CC circuit? It is also necessary to compare the wear rate of the schemes in question. For how long will the resulting effect last?

Reviewer 2 Report

In this manuscript, the authors reported the analysis of ahdesion performance of goat hoof track shoe pattern. I find this investigation of special interest and highly applicable. I would recommend its publication upon the amendment of some minor concerns:

·         Line 48: Which were the optimal running spieed? Please include them.

·         Line 50: Which were the optimal combination parameters?

·         Line 102: What does CT means? The authors must indicate the explanation of the acronyms the first time they appear.

·         Along the manuscript some misspellings can be found. Please, check the body of the text (for example, lines 434-437).

·         Table 7. The “rate of increase” values should be calculated taking AA as the references, but those valures are incorrect. The authors must correct them.

·         Line 471: I guess that the “figure 17” actually refers to figure 16, doesn’t it? Please check all those minor concerns along the whole manuscript.
